# High Risk α-HPV E6 Impairs Translesion Synthesis by Blocking POLη Induction

**DOI:** 10.3390/cancers13010028

**Published:** 2020-12-23

**Authors:** Sebastian O. Wendel, Jazmine A. Snow, Tyler Bastian, Laura Brown, Candy Hernandez, Emily Burghardt, Andrew Kahn, Vaibhav Murthy, Daniel Neill, Zachary C. Smith, Kevin Ault, Ossama Tawfik, Cen Wu, Nicholas A. Wallace

**Affiliations:** 1Division of Biology, Kansas State University, Manhattan, KS 66506, USA; sw87@ksu.edu (S.O.W.); jazasnow@uw.edu (J.A.S.); candyh@ksu.edu (C.H.); emburgha@ksu.edu (E.B.); akahn97@ksu.edu (A.K.); murthy.39@buckeyemail.osu.edu (V.M.); zacharysmith07@ksu.edu (Z.C.S.); 2Department of Statistics, Kansas State University, Manhattan, KS 66506, USA; tbast68@ksu.edu (T.B.); wucen@ksu.edu (C.W.); 3Department of Pathology, University of Kansas Medical Center, Kansas City, KS 66160, USA; laura.brown@ucsf.edu (L.B.); dneill@sbmf.org (D.N.); 4Department of Obstetrics and Gynecology, University of Kansas Medical Center, Kansas City, KS 66160, USA; kault2@kumc.edu; 5MAWD Pathology Group, Lenexa, KS 66215, USA; otawfik@mawdpathology.com

**Keywords:** cervical cancer, translesion synthesis, polymerase eta, Cisplatin, Cisplatin resistance

## Abstract

**Simple Summary:**

Cervical cancers (CaCx) are caused by the expression of human papillomavirus oncogenes (HPV E6 and E7). Here, in vitro assays, computational approaches and immunohistochemical analysis of cervical biopsies show that HPV oncogenes impair translesion synthesis (TLS). This limits the pathway’s ability to prevent replication stress from causing fork collapse and DNA damage. As a result, HPV oncogenes make cells more sensitive to replication stressing agents, such as Cisplatin. Mechanistically, HPV E6 prevents replication stress from triggering the accumulation of a TLS-specific polymerase (POLη). Supplying exogenous POLη to CaCx cells rescues TLS and lowers Cisplatin toxicity.

**Abstract:**

High risk genus α human papillomaviruses (α-HPVs) express two versatile oncogenes (α-HPV E6 and E7) that cause cervical cancer (CaCx) by degrading tumor suppressor proteins (p53 and RB). α-HPV E7 also promotes replication stress and alters DNA damage responses (DDR). The translesion synthesis pathway (TLS) mitigates DNA damage by preventing replication stress from causing replication fork collapse. Computational analysis of gene expression in CaCx transcriptomic datasets identified a frequent increased expression of TLS genes. However, the essential TLS polymerases did not follow this pattern. These data were confirmed with in vitro and ex vivo systems. Further interrogation of TLS, using POLη as a representative TLS polymerase, demonstrated that α-HPV16 E6 blocks TLS polymerase induction by degrading p53. This doomed the pathway, leading to increased replication fork collapse and sensitivity to treatments that cause replication stress (e.g., UV and Cisplatin). This sensitivity could be overcome by the addition of exogenous POLη.

## 1. Introduction

Human papillomaviruses are a large family of small double stranded DNA viruses that are divided into five genera (alpha-, beta-, gamma-, mu- and nu-papillomaviruses) [1]. Among these genera, members of the alpha-papillomavirus genus are most clearly connected to tumorigenesis [2,3]. Alpha-papillomaviruses are designated high-risk or low-risk based on their relative likelihood that their infections cause cancer [4]. High-risk alpha-papillomaviruses cause a subset of head and neck cancers and the vast majority of cervical cancers (CaCx) [5,6]. Further, anal, penile and vulvar carcinomas have been found to be often driven by oncogenic HR-HPV infection [7,8,9]. For simplicity, high risk alpha-papillomavirus will be referred to as HPV moving forward. Each year, HPV infections are responsible for over a half million CaCx that result in over a quarter million deaths [5,6].

HPV expresses two oncogenes (HPV E6 and E7) that effectively reprogram the host cell making the cellular environment more conducive for viral replication [10,11]. HPV E6 and E7 are best known for promoting p53 and Rb degradation, respectively [12,13]. Degrading these tumor suppressors leads to CaCx. However, tumor maintenance requires continual HPV oncogene expression. In addition to degrading p53 and Rb, HPV oncogenes cause replication stress and aberrant activation of some DNA repair responses [14,15,16,17,18].

Replication stress and DNA repair are closely related, as unresolved replication stress results in DNA damage. Translesion synthesis (TLS) helps prevent replication stress from causing DNA damage [19]. Specifically, TLS allows replication forks to bypass DNA lesions that hinder their progression and can cause their collapse. Essential to the process are the ubiquination of PCNA at Lysine 164 (ubPCNA) by RAD6-RAD18 and the switch from a replicative polymerase to a TLS polymerase (POLη/POLH, POLκ/POLK, POLι/POLI, REV1/REV1 or REV3L/POLζ) [20]. Compared to replicative polymerases, TLS polymerases have less restrictive active sites that allow them to incorporate untemplated bases [21]. These bases provide a primer from which replication by replicative polymerases can resume. Although the process is mutagenic, it is less deleterious than the replication fork collapse it prevents.

TLS is activated by Cisplatin-associated replication stress and is a known mediator of Cisplatin sensitivity/resistance [22]. Typically, CaCx can be effectively treated with Cisplatin or other platinum-based drugs that work by causing replication stress [23]. However, Cispaltin-resistance is an unfortunately persistent problem [24]. This manuscript describes how TLS functions in CaCx cells and the extent to which HPV oncogenes hinder the pathway. Data is presented demonstrating that TLS genes are highly expressed in CaCx, with the exception of TLS polymerases. Additional data show that HPV E6 prevents the induction of the best characterized TLS polymerases (POLη) by degrading p53. This causes TLS to fail and contributes to Cisplatin sensitivity. TLS can be rescued by additional POLη, providing resistance to Cisplatin. Finally, examination of The Cancer Genome Atlas CaCx dataset suggests that increased TLS polymerase expression is detrimental to CaCx survival.

## 2. Results

### 2.1. Transcriptomic Analysis Reveals Altered Translesion Synthesis Gene Expression in CaCx

To identify patterns of gene expression alterations in CaCx, a gene ontology analysis was performed on a published set of CaCx gene expression data (GSE 6791) using GOrilla (gene onthology enrichment analysis) software. This approach identified an enrichment of DNA damage response (DDR) genes among the genes with altered expression in CaCx. More specifically, GOrilla identified a significant enrichment of genes in the TLS pathway (*p* < 10^−9^) (Figure 1A). These data were cross-validated using another gene ontology software, GSEA (Gene Set Enrichment Analysis). GSEA found a similar prevalence of DDR and TLS genes among the genes that were differentially expressed in CaCx (Figure 1B). GSEA analysis also identified enrichment of homologous recombination genes as has been previously reported [17].

These analyses showed that alterations in TLS gene expression are common in CaCx, but they did determine if the changes in expression are increases or decreases. They also did not determine the relationship between altered TLS gene expression and CaCx progression. To address this knowledge gap, datasets that linked transcriptomic data with disease progression information were downloaded from the National Center for Biotechnology Information Gene Expression Omnibus (NCBI GEO). Of the seven CaCx datasets available on NCBI GEO, two were excluded because they did not contain stage information [25,26,27,28,29]. The remaining datasets were normalized and merged, yielding transcriptomic data for 262 cervical tissues, divided among tissue with no evidence of disease (NED), premalignant lesions, and CaCx. Figure 1C outlines this process.

An unsupervised cluster analysis then grouped the merged data set by TLS gene expression and CaCx progression (Figure 1D). NED samples differed significantly from both premalignant tissue and CaCx. Among disease tissue, there were three distinct and statistically distinct nodes: 1. premalignant lesions 2. stage I and II cancers and 3. stage III malignancies. Generally, TLS gene expression was increased in premalignant and malignant samples compared to NED tissue. Three distinct groups of gene expression were identified: 1. genes that did not change 2. genes that had decreased expression in premalignant lesions but had no substantial changes in tumors and 3. genes with increased expression in premalignant and malignant tissue. Notably, the expression of four of the five TLS polymerases either did not increase or decreased during disease progression. The complete list of genes tested for expression changes is provided in Appendix A.

### 2.2. In Vitro and Ex Vivo Systems Confirm Dysregulated TLS Gene Expression in CaCx

Then, in vitro and ex vivo systems were used to validate the in silico data. First, TLS gene expression and protein abundance were defined with in vitro models of CaCx. HPV infects keratinocytes, promoting their transformation into CaCx. Primary keratinocyte cell lines can be made from human foreskins and are a commonly used to investigate CaCx and HPV biology. Unmodified HFKs represented untransformed keratinocytes and two CaCx cell lines (SiHa and Hela) represented CaCx. These two CaCx cell lines were chosen because they were derived from cancers caused by the two most common oncogenic HPVs (HPV18 and HPV16). HeLa cells express HPV18 oncogenes, while SiHa cells express HPV16 oncogenes. RT-PCR analysis defined expression of TLS polymerase genes and a representative panel of other TLS genes. This work demonstrated minimal differences in TLS polymerase expression among these three cell lines (Figure 2A). However, the expression of other TLS genes was increased between 2- and 30-fold in a comparison of primary HFKs to the cervical cancer cell lines HeLa and SiHa (Figure 2A). The expression of two genes involved with the activation of TLS (RPA70 and TOPBP1) was also increased [30,31]. Following mentions of Rad6 refer to the protein corresponding to the UBE2B transcript.

Next, immunoblots were used to define TLS protein abundance. We used primary keratinocytes (HFK) as a baseline comparison for primary keratinocytes expressing the HPV16 oncogenes E6 and E7 (HFK E6E7), the HPV^+^ cervical cancer cell lines HeLa and SiHa, and the HPV^-^ cervical cancer cell line C33a. Verification of E6 and E7 expression via degradation of p53 and Rb respectively can be found in Appendix A. These efforts were somewhat restricted by reliable commercial antibody availability, so only two TLS polymerases (POLη and POLκ) were analyzed. There were minimal differences in TLS polymerase abundance (between 1.1 and 1.5 relative fold change) in CaCx cell lines compared to HFKs (Figure 2B,C). However, the abundance of three other TLS proteins (RAD6, RAD18 and PCNA) was increased in CaCx cell lines. These increases ranged from 2- to 4-fold compared to HFKs. TOPBP1 abundance was also increased in CaCx cells. Next, ubPCNA was probed as a marker of TLS activation. It was more abundance in SiHa but not HeLa cells. Increased TLS protein abundance may be a common feature of CaCx as similar increases in TLS protein abundance were found in HPV-positive CaCx cells (HeLa, SiHa) and an HPV-negative CaCx cell line (C33A).

To determine if HPV oncogenes promote the increases in TLS protein, HPV oncogenes (HPV16 E6 and E7) were expressed in HFKs. For simplicity, HPV16 E6 and E7 will be referred to as HPV E6 and E7 moving forward. Immunoblots showing decreased Rb and p53 in primary HFKs expressing HPV16 E6 and E7 (HFK E6E7) were used to confirm expression of the oncogenes (Appendix A). HFK E6E7 had increased PCNA, RAD18, TOPBP1 and ubPCNA abundance (Figure 2B,C). Again, TLS polymerases (POLη and POLκ) were minimally different in HFK E6E7 and HFK cells. While these data support the in silico observations, the immunoblot data did not fully recapitulate our RT-PCR data (Figure 2B,C). There was a greater increase in RAD6 protein than UBE2B mRNA. Further, increases in PCNA expression were more striking at the transcript level than at the protein level.

HPV oncogenes can increase protein abundance by increasing protein stability, so the stability of a subset of TLS proteins (RAD18 and Rad6) and proteins linked to TLS activation (RPA70 and TOPBP1) was defined in HFKs and CaCx cells [14,32]. These proteins were chosen because they were increased at the protein or transcript levels in CaCx cells. To define protein stability, whole cell lysates were collected at intervals after the addition of cycloheximide-containing media (Figure 2D,E). RAD18 and RPA70 were more stable in CaCx cells than in HFKs.

Ex-vivo analysis of 67 premalignant cervical tissues was then used to complement these tissue culture data. The abundance of RAD18, RAD6, PCNA and RPA70 was determined in these tissues by immunohistochemical staining (Figure 3A,B). Although increased abundance was seen for each protein, composite scores only identified significant increases in RAD18 and RPA70 levels between NED and high grade squamous intraepithelial lesions (HSIL). The increased RAD6 detected by immunoblots was not found in this analysis. This staining also provided an indication of the distribution of TLS proteins within cervical tissue. Consistent with TLS’s role of promoting tolerance of replication stress in proliferating cells, TLS proteins were most abundant in basal cells in NED tissue. Likewise, there was increased staining of intermediate and superficial cells in premalignant lesions.

### 2.3. HPV Oncogenes Disrupt TLS Polymerase Eta Induction

To better understand how HPV oncogenes affected TLS gene expression, HPV oncogenes (HPV 16 E6 and E7) were expressed individually and in tandem in HFKs. Immunoblots showing decreased Rb and p53 were used to confirm the individual and combined expression of the oncogenes (Appendix A). At this point, the in vitro analysis of TLS polymerases was restricted to POLη since it has the clearest relationship to tumorigenesis. POLH deletion results in a genetic predisposition to skin cancer [33]. Further, POLH over-expression can drive chemo-resistance [34]. Both HPV oncogenes increased the abundance of some TLS proteins when expressed separately. E6 increased TOPBP1, RPA70 and Rad18 abundance, while E7 increased TLS protein abundance more broadly, including ub-PCNA and POLη abundance (Figure 4A). HPV E7 driven increase of ub-PCNA staining indicates increased TLS activation. RAD6 only increased when both oncogenes were expressed together. While expression of HPV E7 alone increased POLη abundance, the increase was not observed when HPV E7 was expressed along with HPV E6. Thus, while each HPV oncogene can increase the abundance of some TLS protein abundance, HPV E6 attenuates HPV E7-associated increases in POLη abundance when the oncogenes are expressed in tandem.

Next, UV radiation was used to cause replication stress to assess the ability of HPV oncogenes to limit TLS activity. Following UV exposure (5 mJ/cm^2^), whole cell lysates were collected at regular intervals. Demonstrating that these conditions activate TLS, ubPCNA levels were increased in vector control HFKs, HFK E6E7, as well as HeLa cells (Figure 4B). These efforts found no evidence that HPV oncogenes altered TLS activation. After UV, POLη abundance increased in vector control HFKs (Figure 4C,D). However, POLη was not increased in HFK E6E7 or HeLa cells after UV. This motivated an expanded analysis that found the increased POLη abundance was missing when either HPV oncogene was expressed individually. The UV-induced increases in POLη were also absent in SiHa cells exposed to UV. Next, the ability of HPV oncogenes to block POLη accumulation in response to Cisplatin was determined. While Cisplatin caused a dose-dependent increase in POLη abundance in HFKs, this response was absent when HPV oncogenes were expressed in HFKs or in CaCx cells (Figure 4E,F).

Immunofluorescence microscopy was then used to determine if the failure to increase POLη abundance caused defects in TLS. Essers and colleagues demonstrated that PCNA accumulation is associated with TLS initiation and was visible as microscopic foci [35]. As a result, PCNA foci were used as a marker of TLS initiation. PCNA foci were significantly more abundant in HeLa cells at each time point after exposure to 5 mJ/cm^2^ of UV (Figure 5A). This indicates that HPV oncogenes do not hinder the TLS initiation. POLη also forms repair complexes that are detectable by microscopy. After UV, HFKs formed readily visible POLη foci, consistent with progression through the TLS pathway. However, there were minimal POLη foci in UV-exposed HeLa cells (Figure 5B). The lack of POLη foci in HeLa correlated with increased γH2AX foci (a marker of fork collapse) that continued to rise for at least 24 h (Figure 5C) [36]. In contrast, HFKs had a mild induction of γH2AX foci which rapidly resolved. Consistent with these changes being caused by HPV oncogenes, PCNA, POLη and γH2AX foci dynamics were similar among HFK E6 and E7, HeLa and SiHa cells (Figure 5B–E).

To validate that these foci represented TLS repair complexes, their kinetics were compared in HFKs and in HFKs expressing HPV E6 and E7 following UV treatment (Figure 5D,E). As expected, the number of PCNA and POLη foci peak at approximately the same time in HFKs. Further, as these two maxima are reached, γH2AX foci decrease. This is consistent with cells successfully activating TLS, hence protecting themselves from UV-induced replication fork collapse. HPV oncogenes changed these relationships. POLη foci are minimally increased after UV exposure, whereas PCNA and γH2AX rose throughout the time course. These data indicate a situation where the TLS pathway is not functional and as a result UV damage becomes more deleterious. A marker of replication fork stalling (RPA70 foci) was detected to test this interpretation. RPA foci were more prevalent when HPV E6 and E7 are expressed (compare axes in Figure 5D,E). This was true in untreated cells, consistent with published evidence that increased replication stress accompanies HPV E7 expression [18]. These differences became further exaggerated after UV, suggesting that HPV oncogenes prevent TLS from responding to UV-induced replication stress.

### 2.4. HPV E6 Blocks POLη Induction in a p53-Dependent Manner

Despite evidence that both oncogenes hinder POLη induction, published reports more clearly suggest a mechanism for HPV E6 than HPV E7. HPV E6 has a well characterized ability to complex with E6AP and target p53 for degradation [37]. p53 has been shown to bind at the POLH promoter and act as a replication stress-induced transcription factor driving increased POLH expression [38,39,40]. An in silico screen of RNAseq data from 1020 cell lines was used to confirm p53′s ability to regulate POLH expression [41]. Cell lines from this data set were segregated based on relative p53 expression, facilitating a comparison of TLS gene expression between the cell lines with and without low p53 expression. There were 153 cell lines with low p53 expression compared to the other cell lines in the database (z-score < −2). In general, TLS gene expression did not significantly correlate with p53 expression, but TLS polymerases were a notable exception (Figure 6A,B). Three of the five TLS polymerase genes were significantly under expressed in cells with less p53 expression. To determine the extent that these expression changes were specific, the expression of repair genes from five other repair pathways (homologous recombination, non-homologous end joining, base excision repair, Fanconi Anemia repair, and nucleotide excision repair) was evaluated. They are jointly listed as “DNA Repair Genes” in Figure 6A,B and show a slight bias towards over expression when p53 expression is low. Out of the 122 genes queried, POLH had the most significant and most robust decrease in expression in cells with low p53 expression.

These results were validated using isogenic colorectal cancer cells (HCT 116) with and without the p53 gene locus [42]. POLH transcripts were reduced in HCT 116 cells without p53 compared to HCT 116 p53wt cells (Figure 6C). The need for p53 to increase POLη abundance after UV-induced replication stress was also examined. Specifically, proteins were harvested at representative time after UV exposure in both HCT 116 cell lines (Figure 6D,E). p53 and POLη increased in wild type HCT 116. In contrast, immunoblots failed to detect an increase in POLη abundance in HCT 116 cells that lacked p53. Densitometry analysis from these experiments showed a strong linear relationship between POLη and p53 levels. Having seen that p53 was required for the induction of POLη, immunofluorescence microscopy was used to examine p53′s role in POLη foci formation (Figure 6F,G). POLη foci readily appear in wild type HCT 116 in response to UV, but these repair complexes were absent in HCT 116 without p53.

These data establish p53′s role in POLη induction following UV but fall short of demonstrating that HPV E6 prevents POLη induction by reducing p53 abundance in CaCx cell lines. To address this gap, a well-defined dominant negative HPV 16 E6 mutant (HPV16 E6 F47R) was expressed exogenously in HeLa cells. [43]. This mutant binds, but does not facilitate the degradation of p53. This prevents wild type HPV E6 from binding p53. Thus, when transfected in to HeLa cells, HPV16 E6 F47R acts as a specific inhibitor of p53 degradation. Transfection with a concentration gradient of HPV E6 F47R DNA (0–2 μg) into HeLa cells restored p53 in a dose dependent manner (Figure 7A). Empty vector DNA was added to bring each transfection to 2 μg total DNA. Transfection of 2 μg of HPV E6 F47R also caused an increase in POLη abundance (Figure 7B). Further, HPV16 E6 F47R transfection restored POLη foci formation after Cisplatin exposure in HeLa cells (Figure 7C,D).

### 2.5. The Inability to Induce POLη Contributes to Cisplatin Sensitivity in Cervical Cancer Cells

Because of its essential role in TLS, loss of POLη results in sensitivity to crosslinking agents [44]. CaCx cell lines have some POLη. However, they do not increase POLη abundance in response to replication stress, suggesting that they may be more sensitive to crosslinking agents. To test this, HFK and HeLa viability was defined by MTT assay 4, 24 and 48 h after UV exposure (Figure 8A). HeLa cells were only more sensitive to UV than HFKs 24 and 48 h after exposure. This delay is consistent with the time needed for impaired TLS to cause replication forks to collapse. This interpretation is consistent with the immunofluorescence microscopy data shown in Figure 5, where maximum differences in γH2AX staining were not observed until 24 h after UV. As would be expected if disparities in TLS capabilities fueled these differences, HeLa cells were acutely sensitive to low doses of UV where HFK viability was unaltered.

HFK E6E7, HeLa and SiHa cells were also more sensitive to UV and Cisplatin (Figure 8B,C). SiHa was less sensitive to Cisplatin than HeLa cells, as has been previously described [45]. The sensitivity of CaCx cells to crosslinkers extended to Mitomycin C, another clinically relevant compound (Figure 8D). The breadth of this enhanced toxicity lends credence to the hypothesis that TLS is failing by excluding the possibility that CaCx cells have a specific sensitivity to Cisplatin. This hypothesis predicts that the increased sensitivity to Cisplatin seen in HeLa cells is the direct result of these cells not having enough POLη. This was tested in a pair of matched HeLa S3 cell lines, one wild type (HeLa S3 wt) and the other expressing an HA- tagged POLη from a p53-independent promoter (HeLa S3 POLη) [46]. HeLa S3 is a HeLa variant that has been adapted to grow on plates or in suspension. Immunoblots confirmed that HeLa S3 POLη have more POLη protein than HeLA S3 wt (Figure 8E). MTT assays demonstrated that the additional POLη resulted in a consistently decreased Cisplatin sensitivity (Figure 8F).

The increased Cisplatin resistance in HeLa S3 POLη cells could be due to an increased ability to prevent Cisplatin-associated DSBs. Metaphase spreads were used to test this (Figure 9A). DSBs were more common in untreated HeLa S3 WT cells than in HeLa S3 POLη cells (Figure 9A,B). These differences became exaggerated when the cell lines were compared after Cisplatin exposure. Cisplatin caused the chromosomes of HeLa S3 WT cells to lose their structural integrity, a condition known as pulverization. While pulverization occurred in HeLa S3 POLη cells, the frequency was significantly decreased. These data suggest that increased POLη restores TLS function, prevents replication forks from collapsing into DSBs and makes cells less sensitive to Cisplatin. Consistent with this result, HeLa S3 POLη remained sensitive to more direct induction of DSBs by Zeocin, a radiation mimetic that does not require replication fork collapse (Figure 9C).

These data suggest that increased TLS polymerase expression would be deleterious for CaCx that are often treated with Cisplatin. To evaluate this, tumors in The Cancer Genome Atlas (TCGA) CaCx dataset were segregated based on TLS polymerase expression. If a tumor had increased expression of at least one TLS polymerase gene, it was placed in a group termed “TLS POL Expression Increased”. All other tumors were placed in a group called “TLS POL Expression Not Increased”. Increased TLS polymerase expression was defined as having a z-score greater than or equal to 2. There were 58 tumors in the “TLS POL Expression Increased” group and 245 in the “TLS POL Expression Not Increased”. The survival frequency was significantly worse in the “TLS POL Expression Increased” group (*p* = 0.0348, Figure 9D).

## 3. Discussion

Previous investigations of the TLS pathway in context of CaCx has led to conflicting results [47,48,49]. REV3L has been described as both a chemo-resistance factor and as having no significant role in Cisplatin-responsiveness [47,49]. Further, a screen found that knocking down TLS components sensitized CaCx cells to Cisplatin, suggesting the pathway protects CaCx cells from Cisplatin-associated toxicity [48]. Our work helps resolve this ambiguity, showing that CaCx cells are sensitive to Cisplatin at least in part because the TLS pathway is inhibited by an HPV E6-associated POLη insufficiency. Further, we demonstrate that elevated POLH expression can cause Cisplatin resistance in vitro.

Figure 10 offers a visual summary of our interpretation of these data. Briefly, when normal cervical epithelia experience replication stress, it causes both increased expression of TLS genes in general and a p53-driven induction of POLH/POLη (Figure 10A). In these cells, TLS facilitates tolerance of replication stress from Cisplatin, UV and other sources. Cisplatin-associated replication stress would also invoke a p53 response in CaCx, but HPV E6 promotes p53 degradation. This in turn prevents induction of POLH/POLη. The inability to completely activate TLS makes cells sensitive to Cisplatin. These observations are in line with clinical observations that HPV positive CaCx are typically responsive to Cisplatin [50].

Our data demonstrate that both HPV oncogenes can independently prevent the induction of POLη (Figure 4C). Thus, HPV appears to have multiple independent mechanisms of impairing the TLS pathway. Here, we provide mechanistic details of how HPV E6 hinders POLη induction. However, the mechanism by which HPV E7 acts remains poorly defined. Perhaps, the inhibition can be explained by HPV E7-associated inhibition of p53 activity [51]. Our future efforts will determine how HPV E7 hinders POLη induction.

Finally, Cisplatin resistance is a problem shared among many cancer types. It is often hard to predict and without clear clinical recourse [52,53,54,55]. Our data shows that POLη activation (increased abundance and localization) are mediators of Cisplatin sensitivity in vitro. Although we focused on CaCx, POLη likely has a broader influence on Cisplatin resistance. p53 inactivating mutations are ubiquitous in cancer, suggesting that other tumor types have an impaired ability to promote the POLη accumulation. Thus, additional POLH (or other TLS polymerase gene) expression may be a mechanism of Cisplatin resistance shared among divergent tumor types. Indeed, there have been several reports noting the relationship between increased TLS activity and decreased Cisplatin toxicity [22,34,56].

## 4. Materials and Methods

### 4.1. Antibodies and Chemicals

The following primary antibodies were used: GAPDH (Santa Cruz Biotechnologies sc-47724), Ki67 (Abcam ab15580), KIAA0101 (Abcam ab56773), p53 (Calbiochem, OP43-100UG), PCNA (Cell Signaling Technologies 2586S), POLη (Santa Cruz Biotechnologies sc-17770), POLκ (Abcam ab57070), Rad18 (abcam ab57447), Rad6 (abcam ab31917), RPA70 (Cell Signaling Technologies 2267S), TOPBP1 (Santa Cruz Biotechnologies sc-271043), ub-PCNA (Cell Signaling Technologies 13439S), γH2AX (Cell Signaling Technologies 9718S), HA-tag (Cell Signaling 3724S).

All chemicals were purchased from Sigma Aldrich unless stated otherwise.

### 4.2. Pathway-by-Stage and Gene Ontology

NCBI GEO CaCx data sets that included transcriptomic and stage information were identified. Gene expression was normalized using feature scaling to allow inter-data set comparison. The Kyoto Encyclopedia of Genes and Genomes (KEGG) and AmiGO were used to identify gene subsets specific to the following pathways: Replication, TLS, Steroid synthesis [Replication KEGG Pathway: ko03030 [57]; TLS GO-Term: GO:0019985 [58]; Steroid KEGG Pathway: map00140 [59]]. The TLS heatmap and boxplots of individual gene expression by cancer stage were created with the ggplot2 package in the statistical analysis software R and p-values were calculated using students t-test. Fold-change and p-value data available for GEO data set GSE6791 was rank ordered using an established approach for gene ontology analysis [60] that was conducted using both the GOrilla online tool as well as Gene Set Enrichment Analysis software [61,62,63,64]. A threshold of *p* < 10^−5^ was chosen for all gene onthology analyses.

### 4.3. Cbioportal Analysis of TCGA Data

The web-based software available at www.cbioportal.org was used to calculate median survival and to generate Kaplan-Meier Curves of the CaCx samples in TCGA [65,66,67].

### 4.4. Quantitative RT-PCR

mRNA was TRIzol (Ambion) extracted from ~70% confluent HFK, SiHa, and Hela cells, before purification and concentration (Zymo Research). cDNA was synthesized with the iScript cDNA synthesis kit (Bio-Rad). RT-PCR was then performed using an ABI 9700 sequence detection system (Applied Biosystems). The following TaqMan primer/probes (ThermoFisher) were used: ACTB (Hs01060665_g1), PCNA (Hs00427214_g1), RPA70 (Hs00161419_m1), POLK (Hs00211965_m1), POLH (Hs00197814_m1), UBE2B (Hs00163311_m1), RAD18 (Hs00892551_m1), TOPBP1 (Hs00199775_m1), REV3L (Hs00161301_m1), REV1 (Hs01019768_m1), POLI (Hs00969214_m1).

### 4.5. UV-Timeseries and POLη Induction

Cells that were part of UV-time series experiments were exposed to 5 mJ/cm^2^ UV-C at appropriate time points prior to lysate generation.

### 4.6. HeLa/SiHa Nucleoside Addition Experiment

Cells were grown for 3 days in media supplemented with 10 mg/L nucleoside stock solution (Biological Industries Israel Beit-Haemek Ltd. 01-343-1D) before protein lysates were harvested. 

### 4.7. TLS Protein Stability

Protein lysates were collected after indicated length of 100µg/mL cycloheximide exposure and then analyzed by immunoblot. 

### 4.8. Cell Culture

Following commercial cell lines were used:HeLa (ATCC^®^ CCL-2™)SiHa (ATCC^®^ HTB-35™)C-33 A [c-33a] (ATCC^®^ HTB-31™)

Following cell lines were donated and thoroughly characterized by other labs:HeLa S3 WT & HeLa S3 POLη [46]HCT WT & HCT p53-/- [68]

Following cell lines were derived in house from anonymous medical waste:HFK

Cell lines were either grown in DMEM from Gibco supplemented with 10% FBS from VDR Life Sciences (HeLa, C33a HeLa S3 WT, HeLa S3 POLη, SiHa, HCT WT, HCT p53 -/-) or maintained in EpiLife (Gibco) supplemented with HKGS from Gibco (HFKs).

### 4.9. UV and DNA-Damaging Agent Sensitivity

8000 cells/well were seeded on a 96-well plate (Cellstar) and grown for 24 h. UV or Cisplatin dose series treatment was applied 24 h after seeding. Cisplatin was removed after 24 h of treatment via medium change. 48 h after treatment, 10 µl/well of MTT solution (10 mg/mL) was added for 24 h. Subsequently, wells were incubated with 100 µl solubilization solution for 24 h and the optical density measured at 640 nm. Doses for UV with HFK, SiHa, HeLa: 0, 2.5, 5, 7.5, 10, 15, 20, 25, 30, 40, 50, 100 mJ. Doses for Cisplatin with HFK, SiHa, HeLa, HeLa WT and HeLa POLη: 0, 20, 25, 30, 35, 40, 45, 50, 60, 70, 80, 100 µM. Doses for Mitomycin with HFK, SiHa, HeLa: 0, 0.0125, 0.0625, 0.125, 0.1875, 0.25, 0.5, 1, 2.5, 7.5, 10 µM. Doses for zeocin with HeLa WT and HeLa POLη: 0, 1, 5, 10, 20, 30, 40, 60, 80, 100 µg/mL

### 4.10. Damaged Chromosomes Detection by Metaphase Spread

HeLa WT or HeLa POLη cells were grown on 10 cm^2^ plates to 40% confluency and treated with 40 µM Cisplatin for 24 h. Cells were given a 48 h recovery period and damaged chromosomes were detected as previously described [69].

### 4.11. Immunofluorescence Microscopy

Cells were seeded and grown for 24 h on glass bottom 96 well plates (Cellvis), before 5 mJ/cm^2^ UVC exposure. At the indicated times, they were washed and fixed with 4% paraformaldehyde. A 0.1% solution of TritonX in PBS was used to permeabilize the cells. They were blocked with 3% bovine serum albumin in PBS for 30 min. Samples were then incubated with indicated primary and appropriate secondary antibody. Images were taken with a Carl Zeiss 700 inverted microscope and a 40× (1.4 NA Oil) objective. The nucleus was stained using DAPI (300 nM). Foci analysis was conducted using ImageJ as described in [70].

### 4.12. Pathology Analysis

Pathology cases were selected from University of Kansas archives and formalin-fixed, paraffin embedded tissue blocks containing the most representative areas were selected for IHC. The following IHC antibodies were used: RAD6, RAD18, RPA70 (Abcam), and PCNA (Epitomics, Burlingame, CA, USA). Procedures were performed at room temperature using the Biocare IntelliPath autostainer. Epitope retrieval was by Biocare Borg Decloaker (BRCA-1) and citrate with pH 6 (PCNA, RAD6, RAD18, RPA70). Titers used were 1:500 (BRCA-1), 1:600 (PCNA), 1:3000 (RAD6), 1:300 (RAD18), and 1:800 (RPA70), with an incubation of 30 min for all antibodies. Detection was with Dako Evision FLEX HRP (BRCA-1), Biocare Mach 2 Rabbit HRP-polymer (PCNA, RAD6, RPA70), and Dako Envision+ LP, Mouse (RAD18). Slides were counterstained with hematoxylin and permanently mounted. Whole slide images were then analyzed by digital image analysis using Aperio (Leica Biosystems), and results were documented as an expression score incorporating both total percent positivity and intensity of staining [(percent positivity x staining intensity)/100]. Statistical analysis (one-way ANOVA and Tukey-Kramer test) was performed using Microsoft Excel with Real Statistics Resource pack and an alpha level of 0.05.

### 4.13. Statistical Analysis

Two-sided student tests were performed for all data with the exception of the Kaplan Meier curves when LogRank test were used.

### 4.14. E6 F47R Transient Transfections

The pcDNA3 16E6 F47R plasmid was a generous donation from Scott Vande Pol. Cells were seeded at a density of 200,000 cells per well in 4 wells of a 6-well plate. After 24 h the wells were transfected in sequence with 2 µg of DNA, with 0 µg, 0.01 µg, 0.5 µg and 2 µg of the 16E6 F47R plasmid and corresponding values of pcDNA3 vector control. The transfection was carried out according to the Turbofect transfection protocol. Lysates were harvested and analyzed via western blot after a 28 h incubation period. The experiment was repeated a minimum of three times.

For POLη foci analysis via IF, HeLa cells were seeded onto edged 22 mm square coverslips placed in the wells of 6-well plates. Cells were transfected with 2 µg of pcDNA3 vector control or 16E6F47R plasmid according to the Turbofect protocol. After 24 h cells were challenged with Cisplatin for 2 h, fixed, stained and analyzed as described above. The experiment was repeated a minimum of three times.

For POLη and p53 abundance analysis via immunoblot in U2OS cells, cells were seeded at a density of 200,000 cells per well of a 6-well plate. After 24 h the wells were transfected in sequence with 2 µg of DNA (HPV 16E6 wt or HPV 16E6 F47R expression plasmid). The transfection was carried out according to the Turbofect transfection protocol. Lysates were harvested and analyzed via western blot after a 28h incubation period. The experiment was repeated three times

## 5. Conclusions

Our data show that HPV oncogenes hinder the TLS response to replication stress by limiting POLη activation (increased protein abundance and foci formation). HPV E6 prevents these responses by promoting p53 degradation. The inability to induce TLS leads to sensitivity to replication stressors (UV, Cisplatin, etc.) that can be overcome by exogenous expression of POLH.

## Figures and Tables

**Figure 1 cancers-13-00028-f001:**
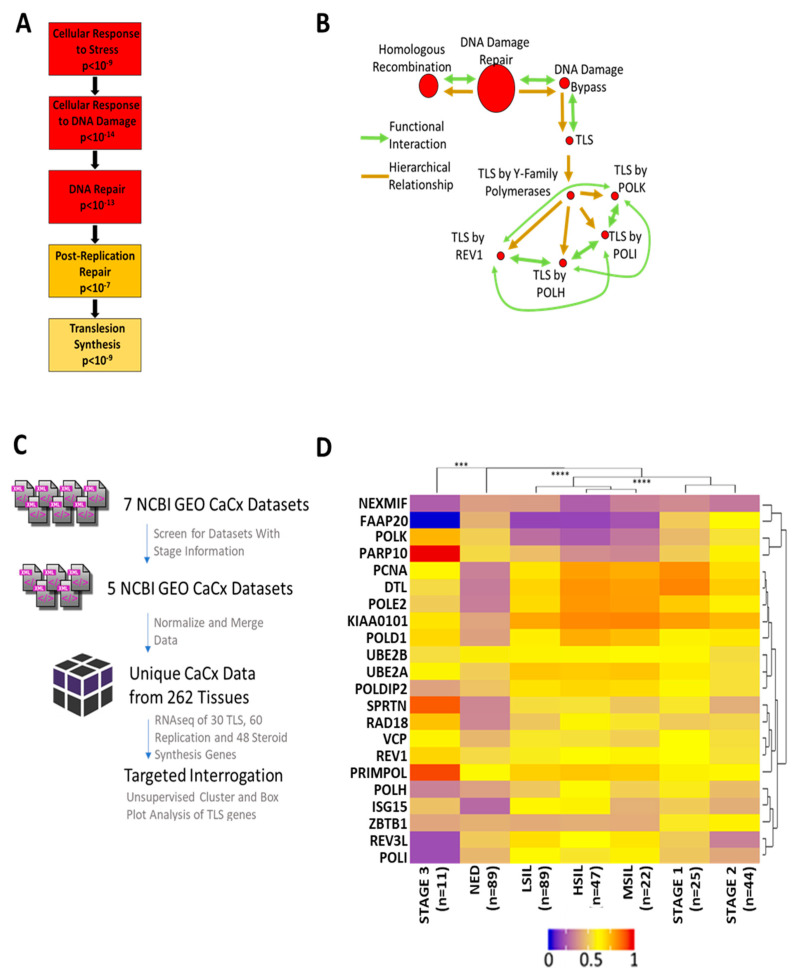
Computational Analysis of Cervical Cancer Indicates a Dysregulation of TLS Gene Expression. (A) Gene ontology (GO) via GOrilla. Boxes show cellular functions in hierarchical order. Boxes descend from general to specific functions. Darker colors indicate greater enrichment of altered gene expression. (B) GO analysis via GSEA. Lines show hierarchical (gold) and functional (green) relationships. Arrowheads indicate directionality. Pathways are shown as red circles. Larger circles mark broader groupings. (C) Flowchart of data-collection, data-formatting and data-analysis. (D) Heatmap of normalized relative TLS gene expression in CaCx. Colder colors indicate lower relative expression. Warmer colors denote higher relative expression. Unsupervised clustering dendrograms are shown along the x and y axes (*** denotes *p* < 0.001, **** denotes *p* < 0.0001; student’s *t*-test).

**Figure 2 cancers-13-00028-f002:**
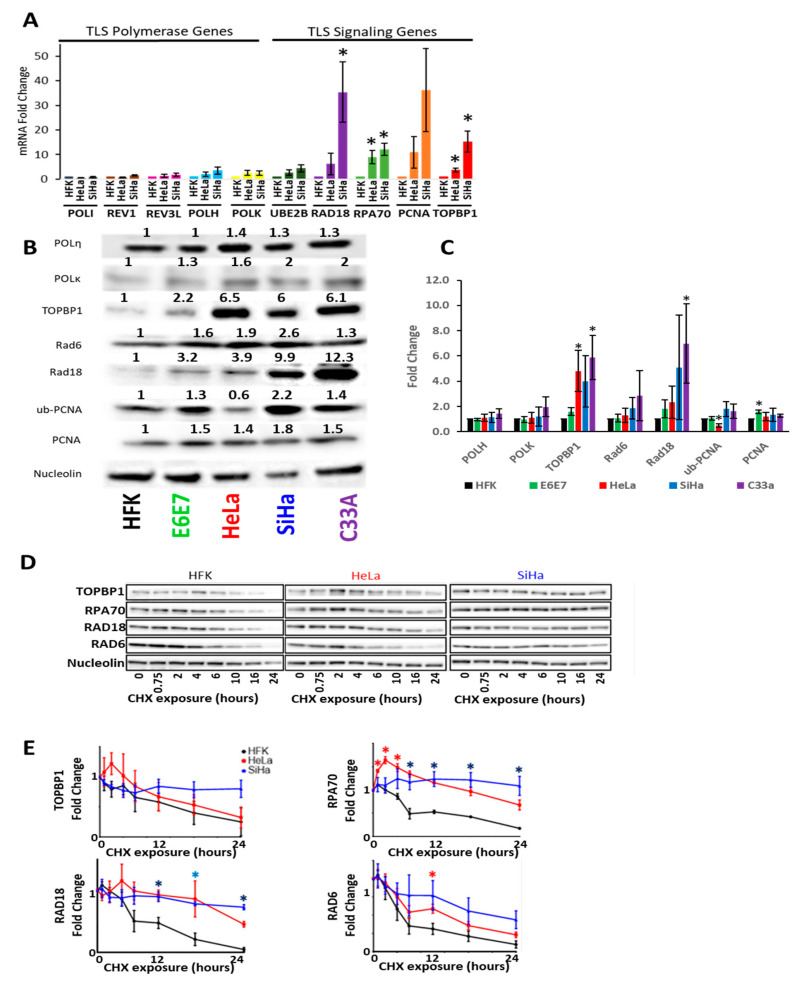
Cervical Cancer Cell Lines Have Increased TLS Gene Expression and Protein Stability with the Exception of TLS Polymerases. (**A**) Average relative mRNA transcripts of TLS polymerase (left) and TLS signaling (right) genes in HFKs and CaCx cell lines. *n* = 3. (**B**) Representative immunoblot of CaCx cell lines, HFKs and HFKs expressing HPV E6E7 (E6E7). Original images can be found in the Appendix A. (**C**) Densitometry of proteins and cells examined in B. *n* ≥ 3. Bars indicate mean densitometry. Error bars indicate SEMs. * denotes *p* < 0.05, students *t*-test (**D**) Representative immunoblots following at noted times after cycloheximide treatment and (**E**) corresponding densitometry. *p*-Value color corresponds to the cell line that is significantly different. *n* = 3. Fold changes were calculated after correcting for loading and normalizing all of the relevant data to mock treated. Red asterisks indicate points were HeLa data was statistically different from HFKs. Blue asterisks indicate points were SiHa data were statistically different from HFKs. Purple asterisks indicate points were HeLa and Siha data were statistically different from HFKs. * denotes *p* < 0.05, student’s *t*-test.

**Figure 3 cancers-13-00028-f003:**
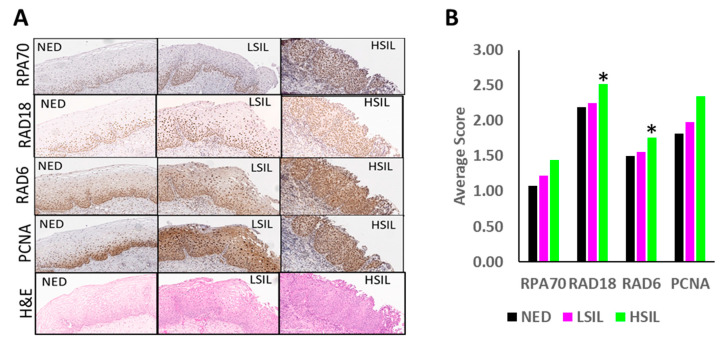
Histology of Premalignant Patient Cervical Biopsies Stained for Markers of Replication, Replication Stress and TLS Activation. (**A**) Representative histology images of H&E, RPA70, RAD6, PCNA and RAD18 staining for NED, LSIL and HSIL. (**B**) Pathology scores of TLS proteins in premalignant lesions. Bars represent NED (black, *n* = 26), LSIL (pink, *n* = 20) and HSIL (green, *n* = 21), * denotes *p* < 0.05, student’s *t*-test.

**Figure 4 cancers-13-00028-f004:**
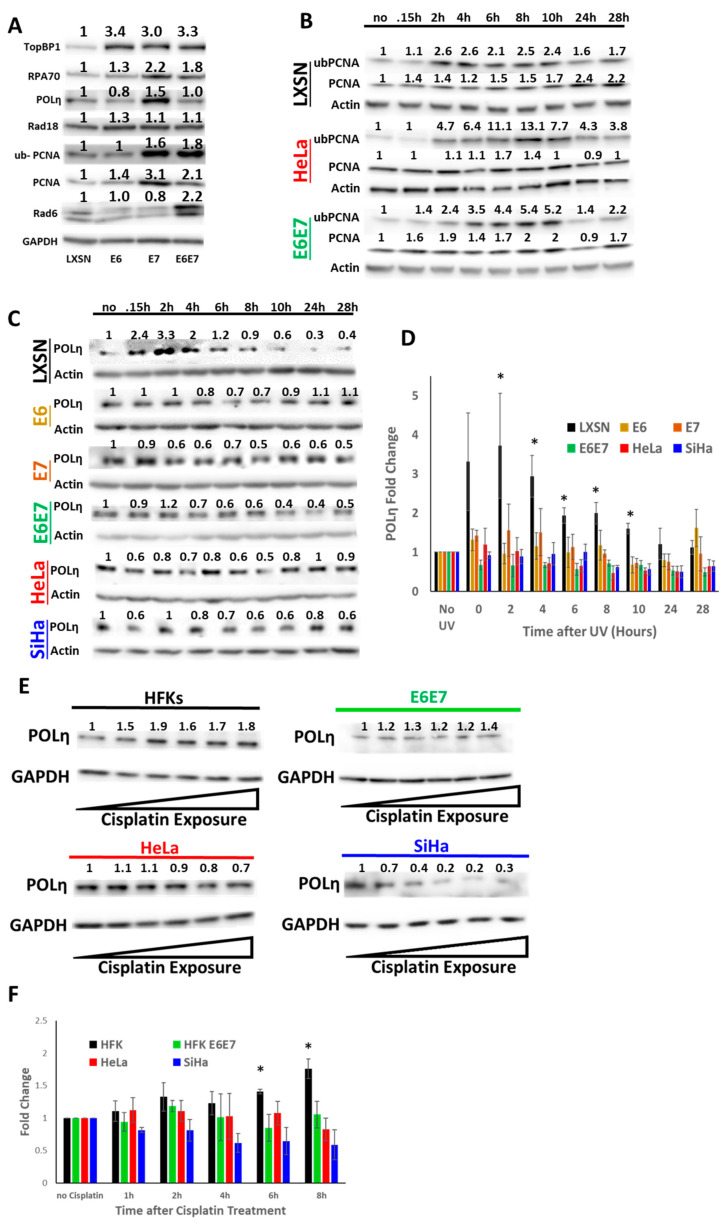
HPV Oncogenes Impair the TLS Response to UV. (**A**) Representative immunoblots of TLS proteins in HFKs expressing HPV oncogenes separately or in combination. Numbers above the proteins represent quantification by densitometry normalized to loading control (GAPDH). Original images can be found in the Appendix A. (**B**) Representative immunoblots of ubPCNA after UV in LXSN (black), HeLa (red) and E6E7 (green). Actin serves as loading control, (*n* = 3). Original images can be found in the Appendix A. (**C**) Representative immunoblots of POLη after UV in LXSN (black), HFKs expressing E6 (yellow), E7 (brown) or E6E7 (green), HeLa (red) and SiHa (blue). Actin serves as loading control. *n* = 3. Original images can be found in the Appendix A. (**D**) Densitometry of POLη corresponding to C (* denotes *p* < 0.05, students t-test). (**E**) Representative immunoblots of POLη after Cisplatin treatment in HFKs, HFK E6E7, SiHa and HeLa cells. Triangles indicate increasing Cisplatin exposure from left to right. Original images can be found in the Appendix A. (**F**) Densitometry of POLη corresponding to E. For all, bars represent means, error bars represent SEMs, and asterisks denote statistical significance compared to untreated control (* denotes *p* < 0.05, student’s *t*-test).

**Figure 5 cancers-13-00028-f005:**
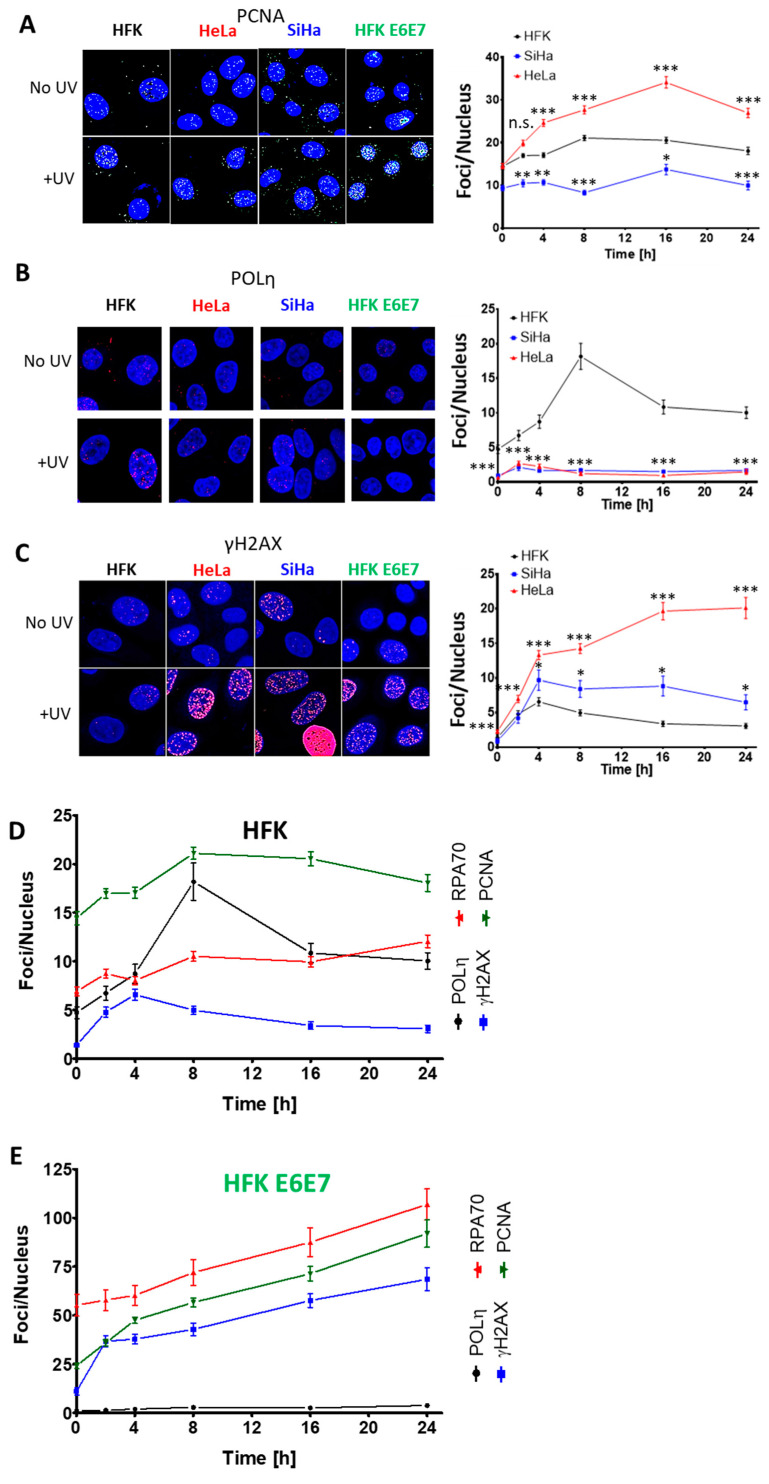
HPV Oncogenes Impair the TLS Response to UV. Representative immunofluorescence microscopy images and average foci per cell of (**A**) PCNA (images at 16 h post UV), (**B**) POLη (images at 8 h post UV) and (**C**) γH2AX (images at 24 h post UV) in HFKs (LXSN), HFK E6E7, SiHa and HeLa cell lines with and without UV exposure. Black data points and lines mark HFKs. Red data points and lines denote HeLa cells. (**D**–**E**) Temporal comparison of POLη (black), RPA70 (red), PCNA (green) and γH2AX foci (blue) after UV treatment for (**D**) HFK and (**E**) HFK E6E7 cells. All data are mean ± SEM. *p*-Value annotation: * denotes *p* < 0.05, *** *p* < 0.001, student’s *t*-test. At least 429 nuclei were counted over three (PCNA) or four (γH2AX, RPA70, POLη) independent experiments.

**Figure 6 cancers-13-00028-f006:**
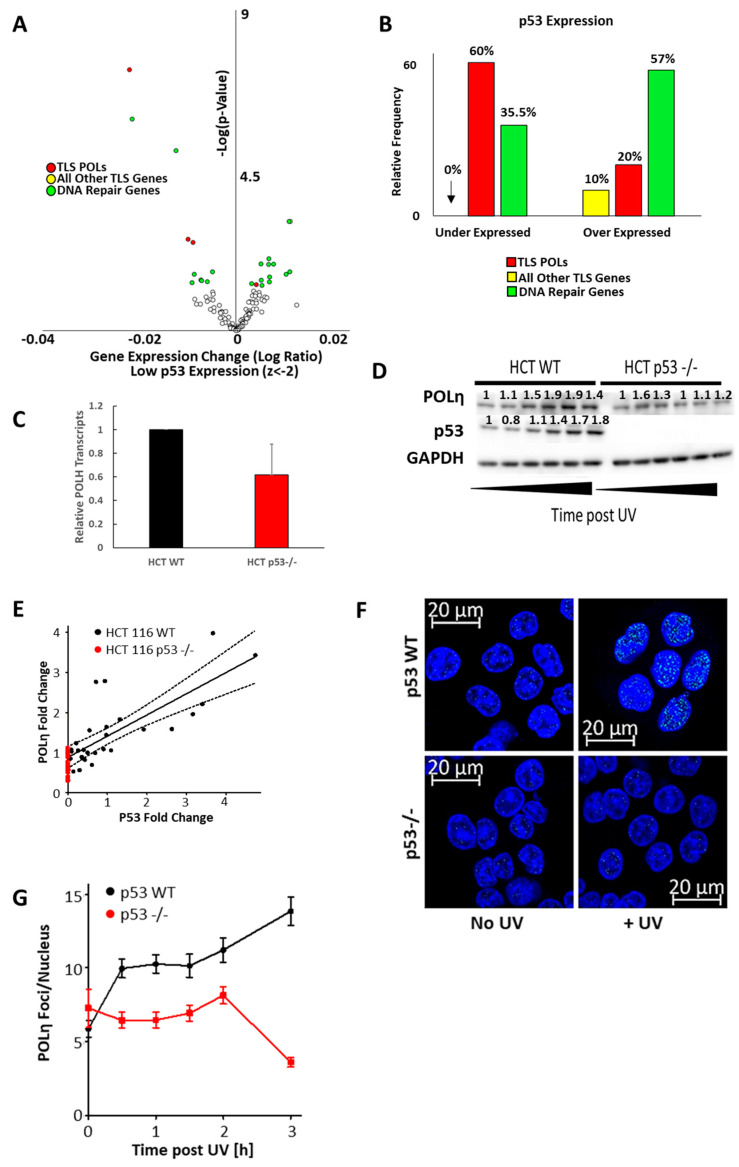
p53 is Required for POLη Induction. (**A**) Volcano plot showing mRNA expression changes of TLS polymerases (red), all other TLS genes (yellow) and other DNA repair genes, when 1020 cell were segregated based on having or not having low p53 expression (z-score < −2). Unfilled circles represent data points that were not statistically significant (*p*-value < 0.05). (**B**) Frequency of TLS polymerase genes (red), all other TLS genes (yellow) and DNA repair genes (green) with significantly (*p* < 0.05) increased and decreased expression. (**C**) POLH transcripts measured by rtPCR in HCT 116 p53wt and HCT 116 p53 -/- cells normalized to ACTB expression as a loading control. (**D**) Representative immunoblots of HCT 116 p53wt and HCT 116 p53 -/- cell lines after UV. (**E**) Graph of the linear relationship between p53 and POLη abundance in HCT 116 p53wt (black) and HCT 116 p53 -/- (red) cells. The solid black line marks the regression curve (*p* < 0.0001) and dotted lines mark the 95% CI. *n* = 5. Original images can be found in the Appendix A. (**F**) Representative image of POLη foci in HCT 116 p53wt and HCT 116 p53 -/- cells with and without UV. (**G**) Graph shows POLη foci per nucleus after UV for HCT 116 p53wt (black) and HCT 116 p53 -/- (red) cells. Over 450 nuclei were examined for each cell line over three independent experiments.

**Figure 7 cancers-13-00028-f007:**
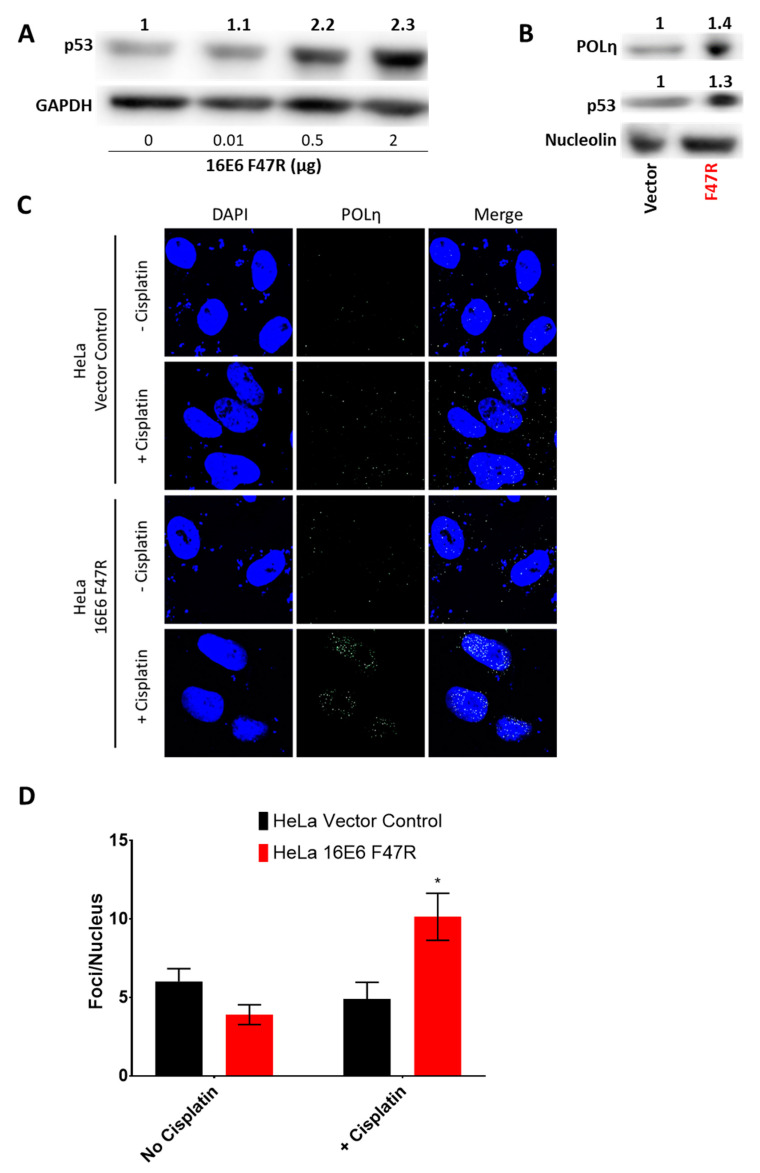
HPV E6′s Degradation of p53 is Necessary to Block the Induction of POLη Foci. (**A**) Representative immunoblot of HeLa cells transfected with increasing amounts of HPV E6 F47R expression vector. Total transfected DNA was equalized by the addition of empty vector DNA. Original images can be found in the Appendix A. (**B**) Representative immunoblot of HeLa cells transfected with either 2 μg of empty vector control (vector) or 2 μg of HPV E6 F47R expression vector (F47R). Original images can be found in the Appendix A. (**C**) Representative immunofluorescence microscopy images of POLη and DAPI stained HeLa cells. These cells were either transfected with an empty vector or with an HPV E6 F47R expression vector. They were also grown in media containing either vehicle or Cisplatin. (**D**) Quantification of POLη foci in at least 42 cells over three independent experiments. Bars are mean ± SEM (* denotes *p* < 0.05 compared to untreated control or empty vector transfected cells, student’s *t*-test).

**Figure 8 cancers-13-00028-f008:**
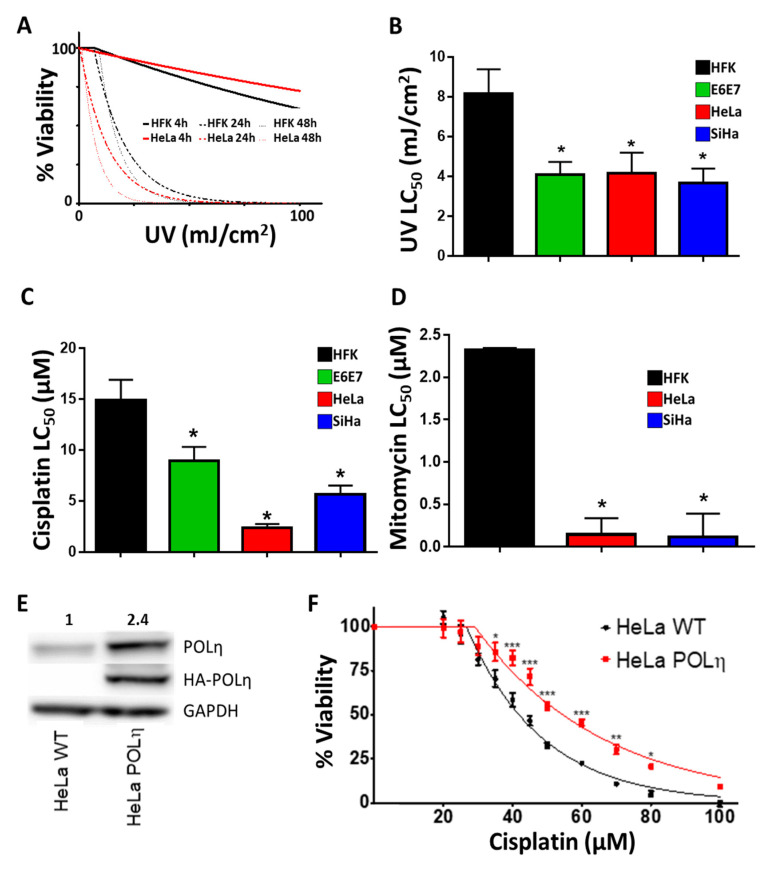
Exogenous POLη Confers Resistance to Cisplatin in HeLa Cells. (**A**). Average relative viability after UV for HFK, SiHa and HeLa cells. Lines show the non-linear regression. Thicker lines indicate earlier time points, *n* = 3. (**B**–**D**) Average LC_50_ for UV, Cisplatin and Mitomycin C in HFK, HFK E6E7, HeLa and SiHa cells. *n* = 3 (**E**) Immunoblot confirming exogenous expression of POLη. Original images can be found in the Appendix A. (**F**) Cisplatin sensitivity of HeLa WT (black) and HeLa POLη (red), *n* = 3. The LC_50_ for cisplatin treated HeLa S3 WT is 41.8 µM, for HeLa S3 POLη 54.6 µM. Bars are mean ± SEM. Asterisks denotes points of statistically significant differences compared to controls. (* denotes *p* < 0.05, ** *p* < 0.01, *** *p* < 0.001, student’s *t*-test).

**Figure 9 cancers-13-00028-f009:**
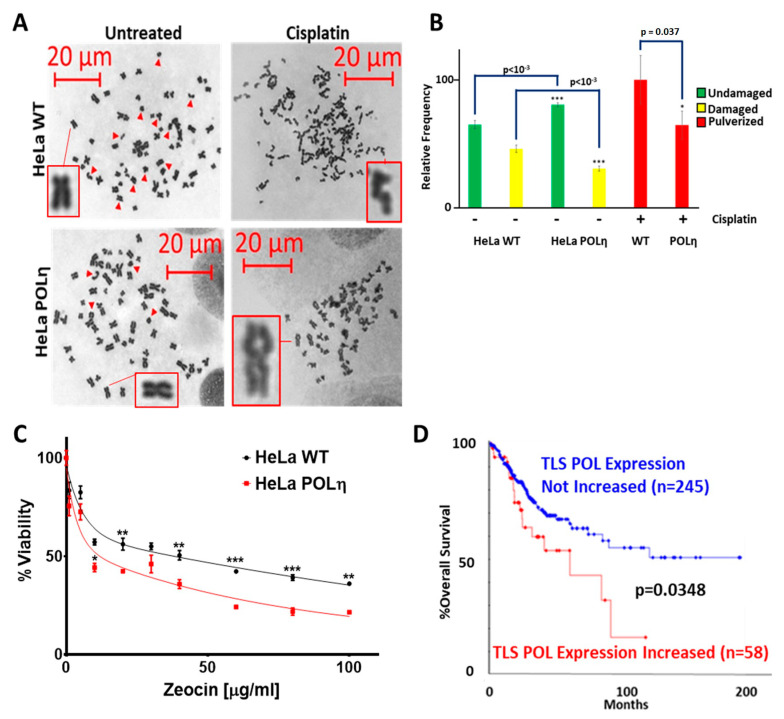
Exogenous POLη Promotes Genome Stability in HeLa S3 Cells. (**A**). Representative images of metaphase spreads. Inserts show magnified images of indicated chromosomes. (**B**) Bar graphs show percentage of undamaged (green), damaged (yellow), and pulverized chromosomes in Hela S3 WT and HeLa S3 POLη cells. 4063 chromosomes were counted from 85 cells over three independent experiments. (**C**) Zeocin sensitivity of HeLa S3 WT (black) and HeLa S3 POLη (red), *n* = 3. All data are mean ± SEM. *p*-Value annotation: * < 0.05, ** <0.01, *** < 0.001, student’s *t*-test. (**D**) Kaplan–Meier (KM) curve of TCGA CaCx survival in individuals over-expressing (red) TLS Polymerases vs. unaltered/decreased TLS polymerase expression (blue). The number of women in each group is shown in parentheses. *p*-Value was obtained via log-rank test.

**Figure 10 cancers-13-00028-f010:**
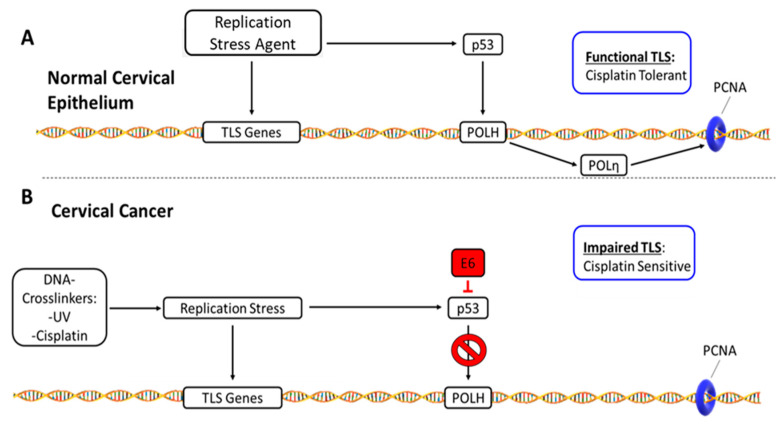
TLS Disruption- and Cisplatin Resistance-Mechanisms in Cervical Cancer. This figure summarizes our data and our interpretation of the data. (**A**) TLS response in normal cervical epithelium: Replication stress induces TLS gene expression and stabilizes p53. Rising p53 results in increased expression of POLH. The resulting increase in POLη facilitates a functional TLS response that promotes tolerance of Cisplatin. (**B**) Disrupted TLS response in HPV oncogene expressing CaCx cells: HPV E6 degrades p53, preventing POLη levels from rising in response to Cisplatin. This impairs TLS, making CaCx cells less capable of mitigating the deleterious effects of Cisplatin-induced replication stress.

## Data Availability

Data available in a publicly accessible repository that does not issue DOIs. Publicly available datasets were analyzed in this study. This data can be found here: [https://www.ncbi.nlm.nih.gov/geo/query/acc.cgi?acc=GSE145976/accession number: GSE145976].

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
