# Peer review of "High Risk α-HPV E6 Impairs Translesion Synthesis by Blocking POLη Induction"

_cancers, 2020, doi:10.3390/cancers13010028_

Round 1
Reviewer 1 Report
This is an excellent study by Wendel et al. showing that HPV E6, via p53 degradation, negatively impacts expression of the TLS polymerase polη. As a result, although replication stress triggers initiation of TLS repair, progress of such repair is halted. While lack of TLS may provide a proliferative advantage to such cancer cells by avoiding collapse of replication forks, additional replication stress imposed by UV or DNA crosslinking agents induces widespread fork collapse and cell death. The manuscript is well written and provides insight into connections between replication stress, TLS repair, and p53 status in viral cancers with possible mechanistic extensions to non-viral cancers.
Major Comment:
While the data with overexpression of E6 and E7 are convincing, the manuscript would be stronger if a few E6 and E7 depletion experiments in HPV-positive cancer lines could be shown.
Minor Comments:
- In Figure 4 and 5 that show data following E6 and E7 expression, immunoblots showing expression of these oncoproteins should be provided.
- In Figure 9D, what fraction of cancers were HPV-positive and was there a difference in survival between HPV-positive and HPV-negative cancers?
- Figure 3 title seems to be incorrect.
- There are some typological errors.
Reviewer 2 Report
The authors of the manuscript entitled” High Risk α-HPV E6 Impairs Translesion Synthesis by Blocking POLη Induction” reported that the expression of α-HPV oncogene E6 impair the translesion synthesis (TLS) (which is a mechanism that helps prevent replication stress from causing DNA damage) in cervical cancer cells by a number of in vitro and in silico assays and IHC analysis. In particular, they demonstrate that HPV E6 prevent replication stress from triggering the accumulation of a TLS-specific polymerase named POLη, while cervical cancer cells resulted to be more sensitive to Cisplatin.
In my opinion the manuscript is well written and organized, results are reported in a comprehensive way and well discussed. I have not major critiques that can prevent the reviewed work from publication.
I have only few minor comments for parts in the manuscript that may deserve improvements.
Line 25: “pathway (TLS)mitigates” --> (TLS) and mitigates should be separated
Line 37. “members of the alpha-papillomavirus genus are most clearly connected to tumorigenesis” --> this sentence is lacking in citations. May I suggest this review PMID: 30133564
Lines 39-40. As correctly reported, “High-risk alpha-papillomaviruses cause a subset of head and nec cancers and the vast majority of cervical cancers (CaCx)”. However, additional important carcinomas have been related to oncogenic HR-HPV (mainly HPV16/18 and, with a lesser extent 11/45) infection. Indeed, penile/anus cancers (PMID: 21543996 and PMID: 33145666) and vulvar carcinoma (PMID: 32266002) have been found to be driven by oncogenic HR-HPV infection. For completeness of information, the tumors/refs mentioned above should be reported in this introductive section.
Line 73 when cited for the first time, DDR genes should be reported as DNA damage response (DDR) genes
Figures. Several WB panels should be improved as the bands are quite difficult to see and/or out of focus. For instance, Fig. 2 B (rad 18), Fig 4 panels C-E (POLη
Reviewer 3 Report
The authors conducted an elegant and well-designed study, which made use of broad spectrum analyses, resulting in important outcomes. Indeed, identification of the HPV E6 oncoprotein action on replication stress through the block of TLS-specific polymerase eta accumulation has important implications on the sensitivity of cervical cancer to replication stressing agents such as cisplatin. I made only a few small suggestions that may slightly improve readability.
Page 1, Simple Summary, line 19: I think that “such as” before Cisplatin is missing.
Paragraph 2.2, page 4, lines 120: it could be specified that the 2 to 30-fold increase of TLS signalling genes occurs in HeLa and SiHa cells with respect to HFK.
Page 5, Figure 2 B-C: it would be preferable to show the proteins of the 2 panels in the same order, perhaps the polymerases first (from top to bottom in B and from left to right in C). Also: it appears that POLk in HeLa cells is the most abundant protein in the immunoblot, which is not apparent from densitometry. Fig. 2B: it is not said that Rad6 is the protein corresponding to the UBE2B transcript.
Comment: honestly, I do not find a good representativeness of the densitometry in C compared to the corresponding protein bands in B. For example, this is evident by comparing Rad 18 in C33A and TopBP1 in Hela, Rad6 and POL eta (in particular in C33A), and also in other cases.
Paragraph 2.2, page 6: as HFK E6E7 cells already appear in Figure 2B, it might be useful to introduce them before the description of the results shown in Figures 2B-C by moving lines 145-148 to line 137, although keeping the analyses separate.
Page 7, Figure 3 A and B: it would be preferable to standardize the order of the panels (from top to bottom) to that of the histograms (from left to right).
Paragraph 2.3, page 7, lines 181-182: it does not seem that the expression of both oncogenes “increased protein abundance when expressed separately”, as pol eta and ub-PCNA are not increased by E6. Lines 187-188: for the same reason, the sentence “with the exception of POL eta, HPV oncogenes individually and at times synergistically increase TLS protein abundance” does not seem appropriate as it applies only to Rad6.
Figure 5A, B, C: the authors chose to show the average foci for HFK and SiHa cells, and not for SiHa and HFK E6E7 cells. Personally, I would prefer to see the trend of all the cells graphed, even if it is said that "the foci dynamics were similar among…” all the cells (line 225). Otherwise, please explain the choice in the figure caption. Figure 5 D-E: it is not possible to deduce either from the figure caption or from the descriptive text on page 9 (lines 226-23) if the kinetics refers to the UV treatment.
Paragraph 2.5, page 11, lines 249-250: this first sentence is the conclusion of the experiments described in the paragraph.
Page 13, line 295: I think the authors mean “degrade” instead of “degradation”.
Page 16, lanes 333-334: the motivation based on different number of genomes between HeLa and SiHa does not apply to UV and mitomycin. However, I do not find the reference appropriate. Figure 8A and 8F: on the y-axis, is “% of Viability” more appropriate than “% Viable”?But it’s up to you. Figure 8 B, C, D: LC50 should be indicated on the Y-axis. Figure 8F, caption: “The LC50 for these cell lines is shown below the graph” is not clear to me.
